# Potential Distribution of Amphibians with Different Habitat Characteristics in Response to Climate Change in South Korea

**DOI:** 10.3390/ani11082185

**Published:** 2021-07-23

**Authors:** Hyun Woo Kim, Pradeep Adhikari, Min Ho Chang, Changwan Seo

**Affiliations:** 1EcoBank Team, National Institute of Ecology, Seocheon-gun, Chungnam 33657, Korea; khw4eco@nie.re.kr; 2Institute of Ecological Phytochemistry, Hankyong National University, Anseong-si 17579, Korea; pdp2042@gmail.com; 3Environmental Impact Assessment Team, National Institute of Ecology, Seocheon-gun, Chungnam 33657, Korea; mhchang@nie.re.kr; 4Division of Ecological Assessment, National Institute of Ecology, Seocheon-gun, Chungnam 33657, Korea

**Keywords:** amphibian species, climate change, habitat characteristics, MaxEnt, species richness

## Abstract

**Simple Summary:**

Amphibian species are one of one of the groups most vulnerable to climate change according to the International Union for Conservation of Nature (IUCN). Limited research has been conducted investigating the effects of climate change on amphibian species in South Korea. In our study, we aimed to predict the impacts of climate change on the distribution of 16 of the 18 species of amphibians currently reported in South Korea. Altogether, 30,281 occurrence points, six bioclimatic variables, and one environmental variable (altitude) were used in modeling. Moreover, we classified 16 Korean amphibians into three groups based on their habitat characteristics: wetland amphibians (Group 1), migrating amphibians (Group 2), and forest-dwelling amphibians (Group 3). Altitude has been predicted to be a major factor in present amphibian distributions in South Korea. In general, our results show that the seven species in Group 1 should be the most resistant to climate change. The five migrating amphibians (Group 2) should decline with preferred habitat reductions. The forest-dwelling amphibian species (Group 3) are the most vulnerable to climate change and their protection requires the immediate implementation of conservation strategies. We will continue to refine our model as it evolves into a useful tool for our endeavor to preserve South Korea’s amphibians as climate change progresses.

**Abstract:**

Amphibian species are highly vulnerable to climate change with significant species decline and extinction predicted worldwide. However, there are very limited studies on amphibians in South Korea. Here, we assessed the potential impacts of climate change on different habitat groups (wetland amphibians, Group 1; migrating amphibians, Group 2; and forest-dwelling amphibians, Group 3) under future climate change and land cover change in South Korea using a maximum entropy modelling approach. Our study revealed that all amphibians would suffer substantial loss of suitable habitats in the future, except *Lithobates catesbeianus*, *Kaloula borealis*, and *Karsenia koreana*. Similarly, species richness for Groups 2 and 3 will decline by 2030, 2050, and 2080. Currently, amphibian species are widely distributed across the country; however, in future, suitable habitats for amphibians would be concentrated along the Baekdudaegan Mountain Range and the southeastern region. Among the three groups, Group 3 amphibians are predicted to be the most vulnerable to climate change; therefore, immediate conservation action is needed to protect them. We expect this study could provide crucial baseline information required for the government to design climate change mitigation strategies for indigenous amphibians.

## 1. Introduction

Global climate was fairly stable before the 20th century, with organisms being well adapted to the typical climate variability. However, organisms have struggled to adapt to rapid climate changes in the late 20th century [1]. These ongoing changes are predicted to negatively impact global species richness and distribution as many organisms change their physiology, phenology, and adaptive capacity [2,3].

Amphibians are excellent indicators of environmental change due to their particular characteristics [4]. Their unshelled eggs, highly permeable skin, and unique biphasic life-cycles make them highly sensitive to changes in thermal and hydric environments [5,6]. Amphibians also require appropriate levels of humidity, temperature, and light, as well as habitats for retreat [7]. Moreover, amphibians play an important role in various food webs as significant predators and food sources of a diverse array of animals [8]. Thus, a decline in amphibian populations negatively impacts ecosystem structure.

Amphibians are one of the most threatened animal groups on Earth [5,9,10]. More than 50% of the 8000 known amphibian species are threatened with extinction [6,11]. The causes for the global decline in their numbers are numerous and complex. These include excessive exploitation of food resources, wildlife trade, habitat destruction and fragmentation caused by land-use change, introduction of alien species such as fish and crayfish, and excessive use of pesticides [11,12,13,14,15].

Climate change is another important factor influencing amphibian decline [6,16]. Climate change severely impacts amphibian populations due to interactions with chytridiomycosis, a fungal disease that is fatal to them [9]; an increase in ultraviolet-B irradiation due to anthropogenic ozone depletion; and the spread of invasive species [11,17]. Therefore, quantifying climate change-driven shifts in species distribution and abundance is necessary for the implementation of effective conservation policies.

Many researchers in South Korea have studied the distribution of amphibian species. Do et al. [18] studied the distribution of eight endangered amphibian and reptile species; identifying altitude as the bioclimatic variable which had the strongest effect on their distributions. Jeong et al. [19] analyzed the microhabitat characteristics of amphibian species living in riparian areas using a habitat suitability model. Jung et al. [20] reported the factors that contributed significantly to the distribution of *Karsenia koreana*. Borzée et al. [21] modeled the current and future predicted distributions of *K. koreana* using a maximum entropy-based (MaxEnt) habitat suitability model which considered four representative concentration pathways (RCPs). However, most of the previous studies have focused on a single species or a limited geographical area. Furthermore, their analyses did not investigate amphibians with similar habitat characteristics.

In this study, we investigated the potential effects of climate change on amphibian species with similar habitat characteristics in South Korea. We used a MaxEnt Model to estimate the suitable habitats for each amphibian species under two climate change scenarios using representative concentration pathways (RCP) 4.5 and 8.5 for the years 2030, 2050, and 2080. We assumed that the species had no dispersal limits, allowing their future distribution to occupy the entire area predicted by the model.

## 2. Materials and Methods

### 2.1. Site Description

The simulation study encompassed the 17 provinces in South Korea, including seven mega-cities and a self-governing city, Sejong (Figure 1). South Korea has a total area of 100,411 km^2^ (Ministry of Land Infrastructure and Transport, 2018). It is largely mountainous, and forests cover over 65% of the land area. The biodiversity of the Korean Peninsula cannot be adequately discussed without mentioning the Baekdudagan Mountain Range [22]. The Baekdudagan Mountain Range is the “backbone” of the Korean Peninsula, stretching from Mt. Jiri in South Korea to Mt. Baekdu in North Korea (Figure 2). It is a well-known biodiversity hotspot across South and North Korea and is relatively untouched by human beings. It harbors approximately one-third of the total flora of the Korean Peninsula, and also diverse fauna including 26% of total bird species, 29% of all mammal species, approximately 30% of the freshwater fish species, and 60% of the amphibian and reptile species in South Korea [22,23]. The country has a temperate climate with four distinct seasons. Moreover, South Korea’s climate is spatially divided into warm temperate (the southern coast and islands), temperate (the central and northern regions), and cold temperate (the high mountains).

The northern region is relatively cold and continental while the southern region is warm and wet. The mean annual precipitation is 1237.4 mm, of which 2/3 falls during the summer season from June to August. The precipitation during the remaining nine months is relatively low. The average annual temperature is 13.2 °C. In the last 30 years, temperature and precipitation have risen by 1.4 °C and 124 mm, respectively, relative to the early 20th century [24].

### 2.2. Species Occurrence Data

According to the Korean Society of Herpetologists, there are 20 amphibian species in South Korea (http://www.krsh.co.kr/html/sub0201.html, accessed on 20 July 2021), and three more *Hynobius* species were reported very recently [25]. Among those 23 amphibians, we obtained occurrence data from the 3rd Natural Ecosystem Survey (Figure 3) [26], and a total of 18 amphibian species were identified [27]. These datasets are available in EcoBank, the first ecological database platform in Korea [28] (Appendix A). However, *Hynobius yangi* and *Dryophytes suweonensis* had less than 10 occurrence points and were therefore excluded from our analysis to obtain accurate model predictions of other species’ distributions. As such, a total of 16 amphibian species and 30,281 species occurrence points (Table 1, Figure 3) were analyzed in this study, and random points were determined from the raster map of South Korea using Arc GIS 10.5.1 (ESRI, Redlands, CA, USA).

### 2.3. Environmental Variables

We investigated 19 bioclimatic variables (Table 2) that influence the distribution of amphibian species. Monthly minimum temperature, maximum temperature, and precipitation data from the Korea Meteorological Administration (KMA) were used to estimate the current and future climate of South Korea. We used Pearson’s correlation between pairs of predictors to select the dominant bioclimatic variables and eliminate any weaker predictors [29,30]. A total of six bioclimate variables including annual mean temperature (Bio1), mean diurnal temperature range (Bio2), isothermality (Bio3), annual precipitation (Bio12), precipitation of the wettest month (Bio13), and precipitation of the driest month (Bio14) were selected for the MaxEnt modeling, similar to the methods of Adhikari et al. [30] and Jeon et al. [31]. Altitude was also weakly correlated with the six bioclimatic variables and was used in the models as an environmental variable. We used the HadGEM3-RA global circulation model via the R Dismo package to simulate future climate change according to the RCP 4.5 and RCP 8.5 scenarios [32,33]. We estimated the current climate conditions by averaging the climatic data from 1950 to 2000 (www.worldclim.org, accessed on 20 July 2021). The 2030 climate conditions were estimated as the average from 2026–2035; the 2050 climate conditions were averaged from 2046–2055; and the 2080 climate conditions were averaged from 2076–2085, similar to the methods of Adhikari et al. [29,30]. The model spatial resolution was 0.01° and approximately 1 km^2^ for all climate and altitude data.

### 2.4. Habitat Characteristics Related to Breeding Behavior

The amphibian species in South Korea are relatively distinct in their habitat characteristics, and we classified them into three groups (Table 3), as detailed by Kim and Song [28] and Lee and Park [34]. The first group (Group 1) were characterized as amphibians that live in low altitude wetlands. These wetland habitats include rice fields [35], riparian zones, and vegetative filter strips alongside streams and lakes [36]. These habitat types mainly occur at lower altitudes, although several wetlands in South Korea also occur at high altitudes [37]. Seven species, including *Dryophytes japonicus*, *Kaloula borealis*, *Glandirana rugosa, Lithobates catesbeianus*, *Pelophylax chosenicus*, *Pelophylax nigromaculatus,* and *Rana coreana*, were grouped into Group 1. The species in Group 2 are migrating amphibians, including *Bombina orientalis*, *Bufo gargarizans*, *Hynobius leechii*, *Hynobius quelpaertensis*, and *Rana uenoi* [38]. They live in terrestrial regions away from waterbodies but migrate to wetlands during breeding periods [38,39,40]. Group 3 consisted of four species, including *Bufo stejnegeri*, *Karsenia koreana*, *Onychodactylus koreanus*, and *Rana huanrenensis*. These species are forest-dwelling amphibians, commonly found in humid regions under rocks on the moist hills of montane woodlands [21,41,42].

### 2.5. Model Development, Evaluation, and Validation

We used the MaxEnt machine learning tool in R (version 1.3.3. https://cran.r-project.org/web/packages/maxent, accessed on 20 July 2021) to predict the current and future distribution of amphibian species in South Korea. During the simulation, 75% of the data was used in the model calibration and 25% was used to test the model’s predictive ability. We performed 10 replications and maintained cross-validation in the replicate runs to guarantee the accuracy of the model.

We evaluated the model’s goodness-of-fit using the area under the curve (AUC) values of the receiver operating characteristic (ROC) curves and the true skill statistic (TSS) [43]. The test data points were used to calculate the AUC and TSS values. The AUC is a threshold-independent technique used to evaluate model performance by differentiating presence from absence [44]. The AUC values varied from 0 to 1, with higher values indicating higher predictive accuracy. Based on the AUC values, the model performance was categorized into poor (0.5–0.6), fair (0.6–0.7), good (0.7–0.8), very good (0.8–0.9), and excellent (0.9–1). The TSS is threshold dependent and measures the model performance by assessing the classification accuracy after selecting a threshold value. The TSS values ranged between −1 and +1. The values “over 0.8” were considered excellent, “from 0.4 to 0.8” were useful, and “less than 0.4” indicated poor model performance [45].

### 2.6. Suitable Habitats and Species Richness

We calculated the suitable habitat areas of amphibian species for the current period, 2030, 2050, and 2080 under RCP 4.5 and 8.5. We summed the raster of each amphibian species to determine the current and potential species richness map. We then overlaid the South Korea shape file on the species richness maps and extracted the multi-values to points using the spatial analyst tool in ArcGIS 10.5.1 (ESRI, Redlands, CA, USA). The extracted richness at different points was used to determine the average and maximum species richness of each amphibian species. For more efficient comparison and explanation, species richness of each amphibian group was categorized into three classes: Low, Mid, and High.

## 3. Results

### 3.1. Model Evaluation and Validation

We established an independent species distribution model (SDM) for the current and future suitable habitats and distributions of the 16 amphibian species (Table 1). We used the AUC and TSS scores for evaluating model performance. The AUC scores ranged from 0.66 for *H. japonica* to 0.99 for *H. quelpaertensis*, and the average was 0.80. The TSS scores varied from 0.53 (*D. japonicus*) to 0.98 (*H. quelpaertensis*), and the average was 0.65. The mean value of AUC indicates good model performance and the mean value of TSS indicates useful model performance [43,44].

Relatively low average AUC values were observed in the species distributed widely across South Korea including *D. japonica*, *H. leechii*, *P*. *nigromaculatus*, and *R*. *uenoi* as the AUC is negatively correlated with widespread species distribution. Conversely, the highest AUC value was for the *H. quelpaertensis* model, which results from the rarity or limited locality of this species [44,46,47]. The occurrence points of *H. quelpaertensis* from the 3rd National Ecosystem Survey of Korea used in this study are restricted to Jeju Island [28]. To compensate the intrinsic prevalence characteristics of the AUC statistic, we used both the AUC and TSS. TSS is independent of prevalence and is less biased than AUC [44].

Even though the AUC and TSS values of some species were relatively low, they are still higher than those from previous studies (i.e., Préau et al. [45]). Therefore, the values are still acceptable to use in discussing current and future potential suitable habitats, distributions, and richness of amphibian species in South Korea.

### 3.2. Variable Contributions in the Model

We selected seven environmental variables for the MaxEnt modeling of 16 amphibian species (Table 3). Altitude showed the highest contribution for 12 of the 16 species, and mean annual temperature (Bio1) was the second-highest contributing factor in Group 1 (4 of 6 species) and Group 3 (3 of 4 species). However, water-related factors, such as Bio12, Bio13, and Bio14, had higher contributions than the temperature-related factors in Group 2. These results seem to relate to habitat characteristics. Species require sufficient water to breed and therefore migrate to waterways, such as wetlands or streams, during the breeding season. However, frequent droughts [48] during their breeding period (February to March for *H. leechii*, *H. quelpaertensis*, *B. gargarizans*, and *R*. *uenoi*; March to April for *B. orientalis*; and May to July for *G. rugosa*) (Chang, 2019. Unpublished data) may negatively impact the habitat suitability for these species.

### 3.3. Climate Change Impacts on Amphibian Species in Each Group

The range of climatically suitable habitats for the 16 amphibian species was estimated using MaxEnt modeling to show the distribution of each species, and calculate the area of potential habitats under current and future climate change conditions (Table 4; Appendix B). Under current climatic conditions, *H. japonica* possessed the largest suitable area (69,191 km^2^), while *H. quelpaertensis* had the smallest area of suitable habitat (1585 km^2^). Under future climatic conditions, *K. koreana* had the largest suitable area for both RCP 4.5 (87,674 km^2^) and RCP 8.5 by 2080 (87,252 km^2^). The smallest suitable areas existed for *B. Stejnegeri* under RCP 4.5 (265 km^2^) and RCP 8.5 (59 km^2^) by 2080. The highest rates of potential habitat expansion by 2080 were identified for *K. borealis* (150.9% increase) under RCP 4.5, and *K. koreana* (91.2% increase) under RCP 8.5. Meanwhile, the lowest rates of potential habitat loss by 2080 were observed in *B. stejnegeri* both under RCP 4.5 (98.5%) and under RCP 8.5 (99.7%). Thus, 14 of the 16 amphibian species under RCP 4.5, and 13 of the 16 under RCP 8.5, followed the general hypothesis that suitable habitat area would decrease by 2080 for both RCP scenarios. The rates of potential habitat loss by 2080 in both RCP scenarios were highest for *B. stejinegeri*. We concluded that *B. stejinegeri* is the most vulnerable amphibian species living in South Korea. However, we could not identify any overall trends when all 16 amphibian species were considered. As such, we present the results for the groups with different habitat characteristics.

### 3.4. Habitat Suitability

#### 3.4.1. Group 1: Wetland Amphibians

Under current climatic conditions, *H. japonica* had the highest suitable habitat area (69,191 km^2^). This is the highest value among the 16 species regardless of group. On the other hand, *P. chosenicus* had the least area of suitable habitat (10,901 km^2^) (Table 4). Under future climatic conditions, *K. borealis* would have the highest suitable habitat area. *P. chosenicus* would have the least area of suitable habitat by 2080 under the RCP 4.5 and RCP 8.5 scenarios. The rates of potential habitat loss were high, ranging from 56.3–81.7% in RCP 4.5 and from 47.4–98.5% in RCP 8.5 by 2080, except for *K. borealis* (−150.9% in RCP 4.5 and −68.1% in RCP 8.5 by 2080) and *L. catesbeianus* (5.1% in RCP 4.5 and −48.3% in RCP 8.5 by 2080). Although the estimated rates and trends of area change were not consistent, the suitable area for amphibian species is predicted to reduce gradually (except for *K. borealis* under both RCP scenarios and *L. catesbeianus* for RCP 8.5).

#### 3.4.2. Group 2: Migrating Amphibians

Under the current climatic conditions, *H. leechi* had the highest area of suitable habitat (68,673 km^2^), while *H. quelpaertensis* had the lowest area of suitable habitat (1585 km^2^) (Table 4). All other Group 2 species have relatively large suitable habitat areas compared to the other groups (Group 1 and Group 3). Under future climatic conditions, *B. gargarizans* had the largest area of suitable habitat, and *H. quelpaertensis* had the least area of suitable habitat by 2030, 2050, and 2080 under RCP 4.5 and RCP 8.5. By 2080, the rate of potential habitat loss for *H. quelpaertensis* was only 4.7% in RCP 4.5 and 23.0% in RCP 8.5. However, in the case of other amphibian species in Group 2, these rates ranged from 67.2% (*B. gargarizans*) to 96.6% (*R. uenoi*) in RCP 4.5 and ranged from 62.7% to 96.5% in RCP 8.5. Under RCP 4.5, the potential suitable areas for all five species consistently decreased by 2030, 2050, and 2080 as we predicted. Any trends in suitable area change were not identified in RCP 8.5, but suitable areas eventually decreased by 2080 for all five species in this group.

#### 3.4.3. Group 3: Forest-Dwelling Amphibians

In the current climatic conditions, *K. koreana* had the highest potential suitable habitat area (45,627 km^2^), and *B. stejnegeri* had the lowest area of suitable habitat (17,160 km^2^) (Table 4). Potential suitable areas for Group 3 amphibian species were generally smaller than those of the other two groups. Interestingly, under future climatic conditions, potential suitable areas for *K. koreana* substantially increased to 92.2% under RCP 4.5, and 91.2% for RCP 8.5 by 2080. Other species in Group 3 experienced large reductions in suitable habitat area. The rate of potential habitat loss by 2080 for Group 3 species (except *K. koreana*) ranged from 97.8% (*R. huanrenensis*) to 98.5% (*B. stejnegeri* and *O. koreanus*) under RCP 4.5, and from 96.0% (*R. huanrenensis*) to 99.7% (*B. stejnegeri*) under RCP 8.5.

### 3.5. Species Richness

#### 3.5.1. Group 1: Wetland Amphibians

The average species richness for Group 1 in South Korea under current climate conditions was 2.60, and the maximum was 6.00 (Table 5). The potential area with low species richness (Low), where 1 or no organisms were found, was 23,489 km^2^; middle species richness (Mid) with 2 to 4 species was 44,823 km^2^; and high species richness (High) with 5 to 7 species was 27,802 km^2^ (Table 6). In the current climate conditions (Figure 4), the potential richness of Group 1 species was highest in the western and southern coastal region along with the provinces, Gyeonggi, Chungcheongnam, Chungcheongbuk, Jeollabuk Jeollanam, Gyeongsangnam, southeastern Gyeongsangbuk, Busan, and Ulsan. Their lowest richness was found in the Baekdudaegan Mountain Range occupying almost all areas of Gangwon Province, along the border between Chungcheongbuk Province and Gyeongsangbuk Province, and the border of Jeollabuk and Gyeongsangnam Province where Mt. Jiri is located.

Under the future climatic conditions, the average species richness of Group 1 was slightly reduced to 2.35 by 2080 (Table 5). Areas of potentially low richness substantially increased from 15,970 km^2^ in 2030 to 34,572 km^2^ in 2050, and to 51,231 km^2^ by 2080 under the RCP 4.5 climate change scenario (Table 6). Under RCP 8.5, similar patterns of gradual increase were maintained but the potential areas of Low species richness in 2080 were larger than those of RCP 4.5. Conversely, potential areas of Mid species richness gradually decreased. Compared to current climate conditions, by 2030 the potential Mid areas increased in both RCP 4.5 (52,094 km^2^) and RCP 8.5 (47,418 km^2^). By 2050, the potential Mid areas had decreased to 48,351 km^2^ and by 2080 they had reached 48,351 km^2^ under RCP 4.5. In RCP 8.5, the potential Mid areas slightly increased by 2050 (48,409 km^2^), and then considerably decreased by 2080 (34,879 km^2^). The patterns of the potential High areas were similar to those of Mid areas, but with substantial variations, especially in 2080 (9672 km^2^ under RCP 4.5, and 8542 km^2^ under RCP 8.5).

The spatial distribution of species richness revealed that the hotspots (areas of high species richness) were restricted to Gyeongsangbuk and Gyeongsangnam Provinces (except for Daegu City and the southern coastal regions), eastern Chungcheongbuk Province, small parts of eastern Gyeonggi Province, and the western part of Gangwon Province under RCP 4.5 (Figure 5a). Under RCP 8.5 (Figure 5b), the average species richness slightly increased to 2.79 by 2080 (Table 5). The main hotspots were similar to those of RCP 4.5, but the areas of distribution extended slightly to adjacent regions. These results indicate that Group 1 amphibian species will respond to climate change better than the other two groups.

#### 3.5.2. Group 2: Migrating Amphibians

Under current climate conditions, the maximum species richness of the second group (Group 2) was 5.00, and the average richness was 3.48 (Table 5). This result is higher than the Group 1 average for seven species. The differences are mainly due to the higher evenness for these species in the current climate conditions (Figure 4). Under current conditions, the potential area of Low (0–1 species) species richness was 22,604 km^2^, Mid (2–3 species) was 36,568 km^2^, and High (4 or more species) was 36,942 km^2^, respectively (Table 6).

Compared with the amphibian species in the other two groups, Group 2 species would be evenly distributed throughout South Korean regions except in the small inland regions of western Gangwon, northern Gyeonggi, eastern Jeollabuk, and Jeollanam Province; and western coastal areas in Gyeonggi, Chungcheongnam, and Jeollanam Province. The lowest richness was found in the Baekdudaegan Mountain Range occupying almost all of Gangwon Province, along the border between Chungcheongbuk and Gyeongsangbuk Provinces, and the border of Jeollabuk and Gyeongsangnam Provinces where Mt. Jiri is located.

Under future climatic conditions, areas of potential Low species richness substantially and constantly increased from 41,113 km^2^ (2030) to 62,642 km^2^ (2050) and then to 89,066 km^2^ (2080) in the RCP 4.5 scenario (Table 6). Under RCP 8.5, the potential Low area in 2050 (59,266 km^2^) is slightly decreased when compared with the Low area in 2030 (63,930 km^2^), but then increases again by 2080 (86,693 km^2^). On the other hand, potential areas of Mid species richness decrease considerably by 2080 for both RCP 4.5 (7098 km^2^) and RCP 8.5 (9359 km^2^) despite great variations in potential area values in 2030 and 2050. The reduction patterns in potential High species richness areas were constant without any unexpected decreases until 2080 (120 km^2^ under RCP 4.5, and 221 km^2^ under RCP 8.5).

Under RCP 4.5 (Figure 5a), the model predicted that the average Group 2 species richness would slightly reduce (to 2.35) by 2080. The hotspots shifted to Gyeongsangbuk and Gyeongsangnam Provinces (except for Daegu city), the southern coastal regions, eastern Chungcheongbuk, and small parts of eastern Gyeonggi and western Gangwon Province. In the RCP 8.5 (Figure 5b) scenario, the average species richness increased slightly to 2.94 by 2080. The main hotspots were similar to those of RCP 4.5, but with the areas slightly extended to adjacent regions. Except for *H. quelpaertensis*, Group 2 species were significantly vulnerable to climate change.

#### 3.5.3. Group 3: Forest-Dwelling Amphibians

Under current climate conditions, the maximum Group 3 species richness was 4.00 and average species richness was 1.25 (Table 5). This result is the lowest among the three groups under current climatic conditions (Figure 4). The potential hotspots for Group 3 species were mainly confined to parts of the mountainous areas in Gangwon Province, and the border of Jeollabuk and Gyeongsangnam Provinces where Jirisan national park is located. Potential suitable habitats of any Group 3 species do not reach Jeju Island, where the Halla Mountain (the highest mountain in South Korea) is located. Under the current climatic conditions, the potential area of Low (0–1 species) species richness was 65,259 km^2^, the largest among the three groups. The potential area of Mid (2–3 species) richness was 25,866 km^2^, and High (4 or more species) was 4989 km^2^, the lowest among the three groups (Table 6).

Under future climatic conditions, potential Low areas substantially and constantly increased from 72,324 km^2^ (2030) to 86,187 km^2^ (2050), and then to 95,317 km^2^ (2080) using the RCP 4.5 scenario (Table 6). Similarly, using RCP 8.5, those values were 86,137 km^2^ in 2030, 92,280 km^2^ in 2050, and 95,001 km^2^ in 2080. On the other hand, potential areas of Mid species richness decrease abruptly and then constantly toward 2080 for both RCP 4.5 (909 km^2^) and RCP 8.5 (1262 km^2^). Reduction in the areas of potential High species richness were constant, and the most drastic, among all three groups under both RCP 4.5 (58 km^2^) and RCP 8.5 (10 km^2^) by 2080.

In the future climatic conditions, potential suitable regions for *K. koreana* were extended all over South Korea, even to Jeju Island, in RCP 4.5 and RCP 8.5. However, regions with high species richness drastically reduced by 2080. Under RCP 4.5 (Figure 5a), the Group 3 hotspots narrowed to the mountainous regions and the eastern Gangwon Province. Under RCP 8.5 (Figure 5b), the main hotspots were similar to RCP 4.5, but the regions were slightly extended into inner Gangwon and southern Gyeongsangbuk Provinces. Based on our results predicting the potential suitable habitats for species under the current and future climate conditions, Group 3 amphibian species (except *K. koreana*) are the most vulnerable to climate change among all three groups.

### 3.6. Overall Species Richness

Changes in potential suitable habitat for the 16 amphibian species can be summarized into three group patterns depending on their habitat characteristics (Table 6). Potential areas of Low species richness constantly increased under RCP 4.5 (73,827 km^2^) and RCP 8.5 (70,203 km^2^) toward 2080. Potential areas of Mid species richness (21,996 km^2^ under RCP 4.5 and 25,945 km^2^ under RCP 8.5) and High species richness (461 km^2^ under RCP 4.5 and 125 km^2^ under RCP 8.5) constantly and drastically decreased toward 2080.

Analyzing the spatial distribution of species richness changes (Figure 5), hotspots could shift into the Baekdudaegan Mountain Range, Daegu basin and its adjacent regions, and other fragmented mountainous areas in the western Gangwon Province, middle Chugcheongbuk Province, and southeastern Chungcheongnam Province.

## 4. Discussion

### 4.1. Current and Future Potential Habitat Suitability

The ranges of climatically suitable habitats and species richness for 16 amphibian species were estimated using MaxEnt modeling under current and future climate change scenarios. The change in the potential habitat suitability for amphibian species depends not only on the climatic variables related to precipitation and temperature used in the MaxEnt model, but also on different non-climatic factors such as overexploitation, habitat loss, and infectious diseases [6,16]. Thus, the suitable habitat for each species may vary under the same climatic conditions in the future.

Our results support previous studies which showed that the potential areas of suitable habitat for almost all the amphibian species in South Korea will decline in the future [11,17]. The rates and extents of habitat decline were not consistent among species but slightly consistent within the groups. The order of PHL was Group 3 > Group 2 > Group 1, indicating that Group 3 species are the most vulnerable, and Group 1 species are the most resistant to climate change. 

Group 1 species distribution could be more sensitive to anthropogenic influence than that of the other two groups. Most of their habitats show possible overlaps with human urban and suburban regions which usually contain water bodies. Thus, interference by urbanization might be one of the most serious factors causing habitat loss and fragmentation [49]. The high-water requirements during migration for breeding are a major limitation for Group 2 species. As such, their habitation may be higher in high rainfall areas compared to areas with low water availability, which constitute a loss of species habitat for this group [38]. The land-use conversion from forest to agricultural land, and the resultant habitat fragmentation, may make forest-dwelling amphibians (Group 3) more vulnerable to climate change [41]. 

On the other hand, climate change may promote the distribution of some species, creating opportunities for ecosystem restoration [50]. In this study, the suitable habitat areas of three species, *L. catesbeianus*, *K. borealis*, and *K. koreana*, were predicted to increase by 2080. 

The American bullfrog (*L. catesbeianus*) is not native to South Korea but has successfully adapted to Korean environment since it was first introduced in 1957. In 1998, it was designated as an ecosystem-disturbing, invasive species. This species has virtually no natural predators and eats insects, fish, snakes, and native frogs [51]. In 2030, the potential area of suitable habitat for the American bullfrog is predicted to substantially increase. Even though the total habitat area of the American bullfrog is predicted to slightly reduce in 2050 and 2080, this species is likely to spread across the country; therefore, a long-term management plan is required [52].

The distribution of the narrow-mouthed toad (*K. borealis*) had increased substantially by 2080. This result may be closely related to precipitation increases under RCP 4.5 and 8.5 [53]; 17.3% and 20.4%, respectively. More research is required to investigate the trends in this species’ future distribution.

The most striking results of this study were those regarding the potential distribution of the Korean crevice salamander (*K. koreana*), the first lungless salamander (Plethodontidae) species discovered in Asia [42]. Under current climate conditions, the potential suitable habitat for *K. koreana* was limited to the middle portion of South Korea. However, this species was predicted to spread widely throughout Korea in 2030, 2050, and 2080 (Figure 5). Borzée et al. [21] predicted a similar trend in the potential habitat distribution of *K. koreana* under climate change using the MaxEnt model with four RCP scenarios and six bioclimatic factors. Even though their projections only included the years 2050 and 2070, they found marked increases in the potential habitat areas under RCP 4.5 in both years. However, under RCP 8.5, there were substantial increases in 2050 followed by substantial decreases in 2070. These contrasting results may have been caused by the use of different bioclimatic variables. They used four temperature-related variables including bio2, bio3, bio4, and bio6, and two precipitation-related variables such as bio14 and bio16. Furthermore, they did not include altitude as an input environmental variable. In our study, altitude was identified as the most influential variable for habitat suitability in amphibians. As such, we suggest that future studies investigate various approaches with diverse bioclimatic input variables and ecological settings to understand the potential current and future distribution of this enigmatic species under climate change.

### 4.2. Potential Species Richness under Current Conditions

The current potential distributions of amphibian species reflect the patterns in the topography of Korea and are represented by the Baekdudaegan Mountain Range [22]. The potential distribution of Group 1 species was concentrated in the southwestern area of South Korea where wetlands are located, such as rice fields, riparian buffers, and vegetative filter strips. Group 2 amphibians were predicted to live throughout the South Korean region. The Baekdudaegan Mountain Range was predicted to be a main hotspot for Group 3 amphibian species.

### 4.3. Potential Species Richness under Future Conditions

As we briefly discussed, by 2080, the potential hotspots for Group 1 and Group 2 species under RCP 4.5 and 8.5 would shift to the Daegu basin, south of Mt. Taebaek in Baekdudaegan Mountain Range. This trend was more evident in Group 2 than in Group 1. This convergence could result from the high precipitation expected in this region in the future. The Korea Meteorological Association (Korea Meteorological Administration 2017) predicted that the rate of precipitation increase in the Daegu region would be higher than the average for the entire South Korean region until the end of the 21st century, but the rate of temperature increase would be lower than the South Korean average. As Group 2 species require a large amount of water during the breeding period, they are predicted to disperse into the areas of higher precipitation.

In the case of Group 3, the potential distributions drastically decreased for three species including *B. stejnegeri*, *O. koreanus*, and *R. huanrenensis*; and their habitats were generally limited to the northeastern region stretching from Mt. Seorak to Mt. Taebaek, in the middle of the Baekdudaegan Mountain Range. Our study suggests Group 3 species were the most vulnerable group to climate change. This finding is consistent with earlier studies investigating the impacts of climate change on high-altitude biodiversity, including amphibians [30,32,54]. However, this is an unexpected result considering the three different habitat characteristics. We assumed that larger areas of future Group 1 habitats were inclined to be overlapped with human activities, compared to the future habitat areas of Groups 2 and 3. In the case of Group 2, they could face water scarcity and road kill problems during the migration period. Therefore, we assumed that Group 1 or Group 2 were more vulnerable than Group 3. Moreover, as we discussed previously, a substantial increase in the potential distribution area of *K. koreana* was predicted. More studies on the various aspects which impact *K. koreana* distribution, including land-use change and habitat fragmentation, are required.

The results of this study may provide a baseline for designing different conservation strategies for amphibians indigenous to South Korea. In 2005, the International Union for Conservation of Nature hosted the Amphibian Conservation Summit and declared the Amphibian Conservation Action Plan (ACAP) [55]. The ACAP included four fundamental actions, first, understanding the causes of decline and extinction of amphibians. Second, ongoing documentation of amphibian diversity and changes therein. Third, development and implementation of a long-term conservation plan. Finally, undertaking emergency responses to urgent crises [55]. We expect that this study will provide baseline information about current and future potential habitats of amphibian species in South Korea, which could be an important reference for designing conservation policies, including in situ and ex situ conservation of threatened species [56]. We predicted that the middle part of the Baekdudaegan Mountain Region and southeastern Gyeongsangbuk Province would be future potential habitats for many species. Therefore, they could be priority regions for establishing amphibian conservation area in future. Policy-makers can use these results to establish the long-term conservation plan in this region, which should include habitat management, reductions in anthropogenic pressure, and controls on hunting. In addition, our study suggests that forest-dwelling amphibians are at high risk under future climate change and this requires species-specific conservation programs for their long-term conservation in South Korea [57].

### 4.4. Limitations and Possible Future Research

Our approach is comprehensive and deals with almost all amphibian species identified from the National Ecosystem Survey in South Korea, even though two species were excluded due to the lack of minimum occurrence points for MaxEnt modeling. However, there are still some limitations in this study. First, we did not include land-use data. The integration of land-use data in SDM may provide invaluable information to evaluate the future distribution changes of amphibians, improve the strength of predicted results, and assist in outlining potential conservation measures [45,58]. Land-use is also closely related to the scale problem. Pixel size in this study was 1 km^2^, but our baseline simulation results using a land-use variable with 1 km^2^ scale were not refined enough to provide any influence on the variations in suitable habitat and species richness of amphibian species. In addition, we assumed unlimited dispersal, though migration rates would assist in more accurately determining future potential distributions and species richness [59]. Consequently, this is the first step towards developing a future research program where we can utilize our results as a base and address the identified limitations.

## 5. Conclusions

We investigated the potential effects of climate change on the distribution of 16 amphibian species with different habitat characteristics in South Korea. These included wetland amphibians (Group 1), migrating amphibians (Group 2), and forest-dwelling amphibians (Group 3). Our results indicated that suitable amphibian habitats will decrease continuously toward 2080 under both RCP 4.5 and 8.5, except for three species; *L. catesbeianus*, *K. borealis*, and *K. koreana*. The future potential habitat expansion of *K. koreana* is the most striking, and immediate further investigation into the various factors affecting this species’ survival is required. The species richness of Group 1 and Group 2 will converge southeastward, around the Daegu basin. This trend was more obvious in Group 2 than in Group 1. The hotspots of Group 3 species (except *K. koreana*) will shift into the middle of the Baekdudaegan Mountain Range, along the eastern coast. Unexpectedly, species in Group 3 were the most vulnerable to climate change among the three groups. Thus, special conservation measures are required. The results of this study provide useful information on how climate change will influence the decline of amphibian species in South Korea based on their habitat characteristics.

## Figures and Tables

**Figure 1 animals-11-02185-f001:**
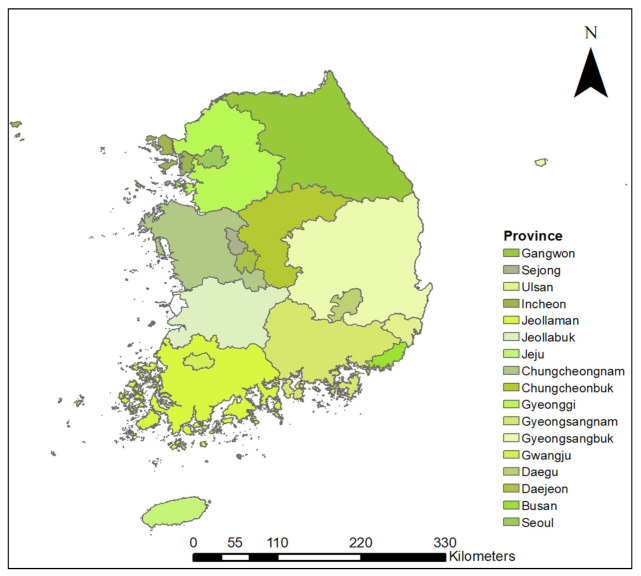
Map of study area. The 17 colors in the map indicate different provinces and mega cities of South Korea.

**Figure 2 animals-11-02185-f002:**
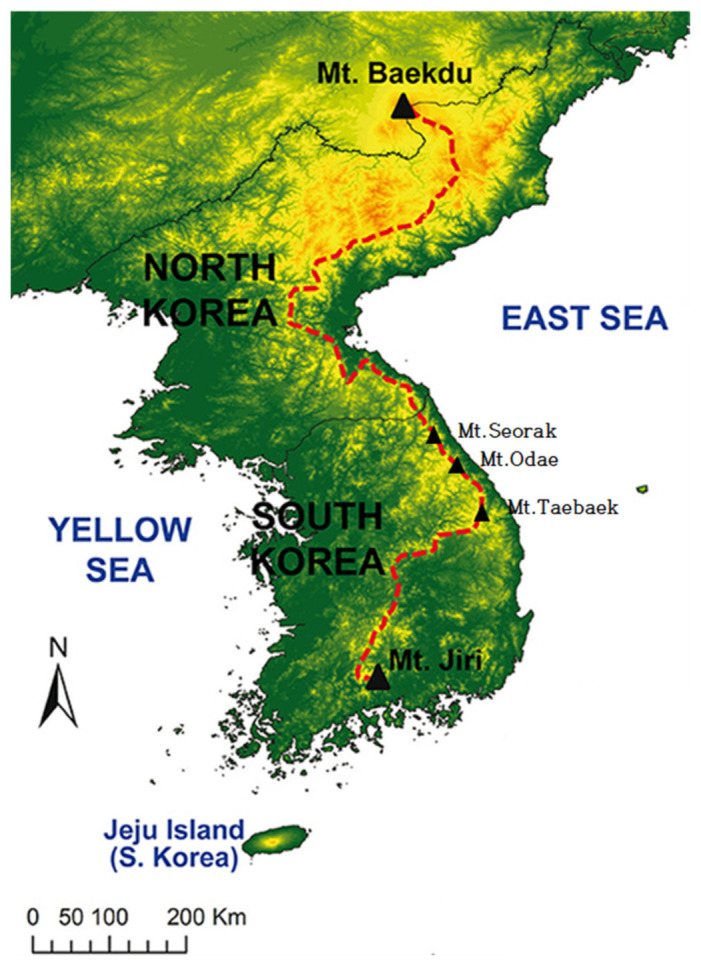
The Baekdudaegan Mountain Region in Korea (Modified from: Chung et al., 2018 [22]). The Baekdudaegan Mountain Region is the “backbone” in Korean Peninsula. It stretches across the Korean peninsula and is a well-known biodiversity hotspot, harboring 1500 plant species (in the South Korean area.

**Figure 3 animals-11-02185-f003:**
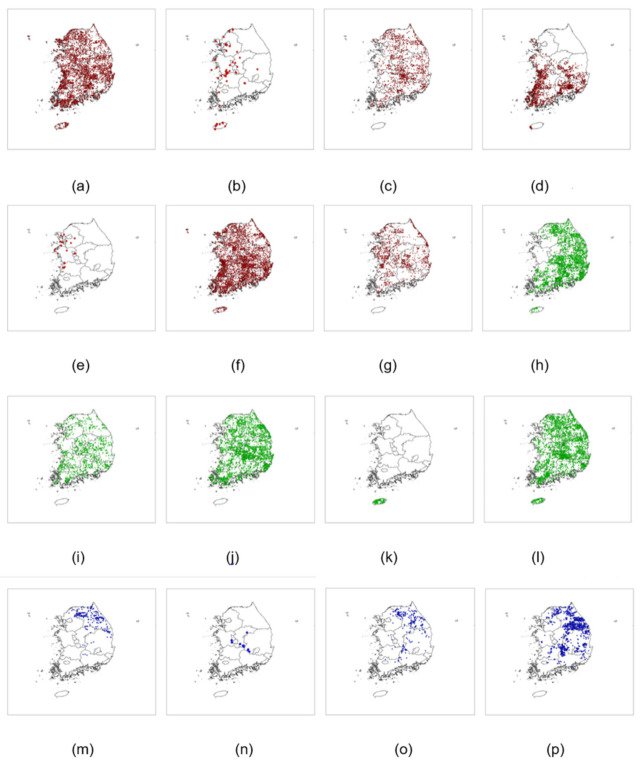
Occurrence points for 16 amphibian species: (**a**) *H. japonica*; (**b**) *K. borealis*; (**c**) *G. rugosa*; (**d**) *L. catesbeianus*; (**e**) *P. chosenicus*; (**f**) *P. nigromaculatus*; (**g**) *R. coreana*; (**h**) *B. orientalis*; (**i**) *B. gargarizans*; (**j**) *H. leechii*; (**k**) *H. qualpartensis*; (**l**) *R. dyboskii*; (**m**) *B. stejnegeri*; (**n**) *K. koreana*; (**o**) *O. koreanus*; (**p**) *R. huanrenensis*.

**Figure 4 animals-11-02185-f004:**
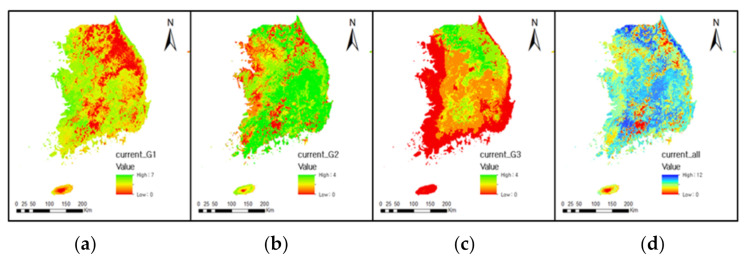
Potential species richness of amphibian species under the current climate. The figures (**a**–**d**) show the richness of amphibians for Group 1, Group 2, Group 3, and overall.

**Figure 5 animals-11-02185-f005:**
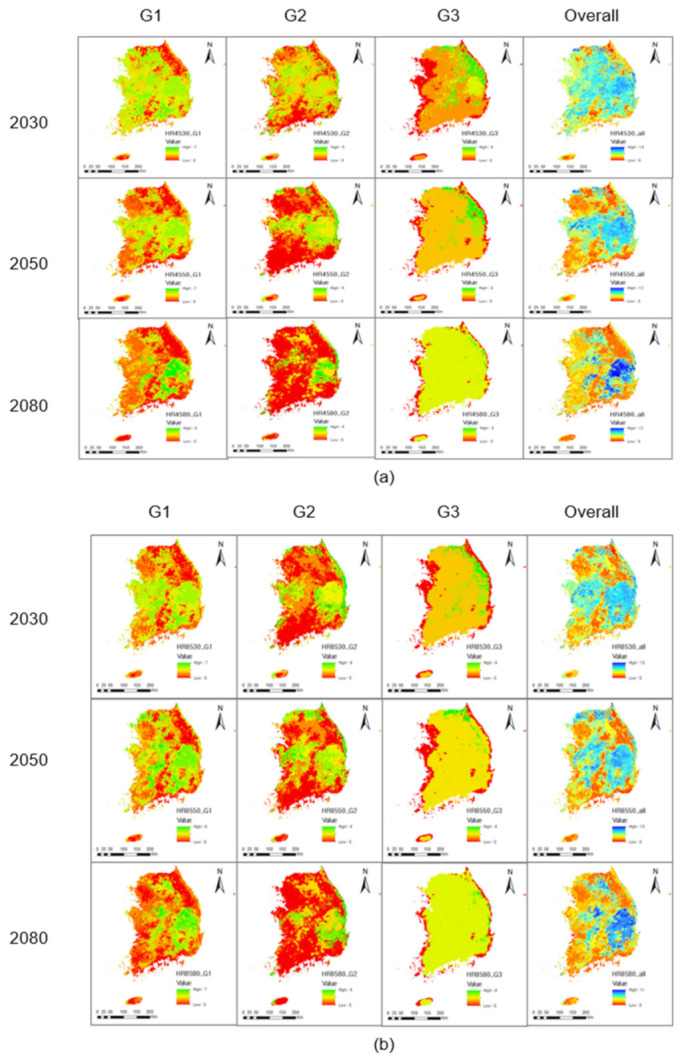
Potential species richness of amphibian under the future climate change: (**a**) RCP 4.5 (**b**) RCP 8.5.

**Table 1 animals-11-02185-t001:** List of amphibian species in South Korea.

Family Name	Scientific Name	Common Name	Presence	AUC	TSS
Bufonidae	*Bufo gargarizans*	Asian toad	1233	0.79	0.59
Bufonidae	*Bufo stejnegeri*	Water toad	233	0.92	0.72
Discoglossidae	*Bombina orientalis*	Oriental fire-bellied toad	3302	0.73	0.54
Hylidae	*Dryophytes japonicus*	Japanes tree frog	4976	0.66	0.53
Hynobiidae	*Hynobius leechii*	Korean salamander	3401	0.77	0.65
Hynobiidae	*Hynobius quelpaertensis*	Jeju salamander	74	0.99	0.98
Hynobiidae	*Onychodactylus koreanus*	Korean clawed salamander	240	0.89	0.66
Microhylidae	*Kaloula borealis*	Narrow-mouthed toad	64	0.87	0.65
Plethodontidae	*Karsenia koreana*	Korean crevice salamander	13	0.82	0.69
Ranidae	*Glandirana rugosa*	Wrinkled frog	1683	0.71	0.61
Ranidae	*Lithobates catesbeianus*	American bullfrog	2527	0.87	0.59
Ranidae	*Pelophylax chosenicus*	Korean golden frog	31	0.97	0.89
Ranidae	*Pelophylax nigromaculatus*	black-spotted pond frog	6314	0.78	0.65
Ranidae	*Rana coreana*	Korean brown frog	1562	0.76	0.57
Ranidae	*Rana uenoi*	Korean large brown frog	3708	0.68	0.55
Ranidae	*Rana huanrenensis*	Huanren brown frog	930	0.86	0.54
Hylidae	*Dryophytes suweonensis* *	Suweon tree frog	Presenc epoints < 10		
Hynobiidae	*Hynobius yangi* *	Kori salamander	Presenc epoints < 10		
Hylidae	*Dryophytes flaviventris* ^a^	Yellow-bellied tree frog	www.krsh.co.kr (accessed on 20 July 2021)		
Hynobiidae	*Hynobius geojeensis* ^a^	Geoje salamander	Borzée and Min, 2021		
Hynobiidae	*Hynobius notialis* ^a^	Southern Korean salamander	Borzée and Min, 2021		
Hynobiidae	*Hynobius perplicatus* ^a^	Cryptic Uiryeong salamander	Borzée and Min, 2021		
Hynobiidae	*Hynobius unisacculus* ^a^	Korean small salamander	www.krsh.co.kr (accessed on 20 July 2021)		

* There are 23 amphibian species currently recorded in South Korea. We targeted 18 species to model, identified from the 3rd National Ecosystem Survey in Korea, but *D. suweonensis* and *H. yangi* were not included in analysis as their number of occurrence points were less than 10. ^a^ Newly reported amphibian species in South Korea. We excluded these species in analysis as their occurrence points are not available on the datasets collected by the 3rd National Ecosystem Survey. (Species in gray shaded area were not included in our simulation).

**Table 2 animals-11-02185-t002:** Bioclimatic and environmental variables used in this study.

Code	Description	Unit	Source
Bio1	Annual mean temperature	°C	KMA
Bio2	Mean diurnal temperature range	°C	KMA
Bio3	Isothermality (BIO2/BIO7) (×100)	Percent	KMA
Bio4	Temperature seasonality	°C	KMA
Bio5	Max temperature of warmest month	°C	KMA
Bio6	Min temperature of coldest month	°C	KMA
Bio7	Temperature annual range	°C	KMA
Bio8	Mean temperature of wettest quarter	°C	KMA
Bio9	Mean temperature of driest quarter	°C	KMA
Bio10	Mean temperature of warmest quarter	°C	KMA
Bio11	Mean temperature of coldest quarter	°C	KMA
Bio12	Annual precipitation	mm	KMA
Bio13	Precipitation of wettest month	mm	KMA
Bio14	Precipitation of driest month	mm	KMA
Bio15	Precipitation seasonality	Fraction	KMA
Bio16	Precipitation of wettest quarter	mm	KMA
Bio17	Precipitation of driest quarter	mm	KMA
Bio18	Precipitation of warmest quarter	mm	KMA
Bio19	Precipitation of coldest quarter	mm	KMA
Dem	Elevation	mm	KMA

KMA: Korea Meteorological Administration.

**Table 3 animals-11-02185-t003:** Estimating the contributions of environmental variables for the modeling of 16 amphibian species among habitat groups.

Groups	Species	Altitude	Bio01	Bio02	Bio03	Bio12	Bio13	Bio14
Group 1	*D. japonicus*	81.4	4.4	3.5	3.7	0.4	6.2	0.3
	*K. borealis*	48.5	7.1	0.3	6.9	16.9	10.6	9.7
	*G. rugosa*	60.1	2.6	6.9	12.7	11.6	1.9	4.3
	*L. catesbeianus*	31.2	46.1	0.8	2.3	2.6	16.4	0.6
	*P. chosenicus*	78.6	10.4	1.4	8.1	0.4	0.3	0.8
	*P. nigromaculatus*	77.4	9.0	3.6	7.3	1.6	0.8	0.2
	* R. coreana*	43.5	25.5	4.1	2.7	13.5	5.3	5.3
Group 2	*B. orientalis*	34.9	8.4	1.6	4.5	2.7	34.1	13.9
	*B. gargarizans*	56.0	9.4	2.0	6.7	10.1	15.5	0.3
	*H. leechii*	44.4	4.6	3.9	1.9	15.3	27.4	2.4
	*H. quelpaertensis*	0.0	6.7	0.3	0.3	60.1	0.3	32.3
	* R. uenoi*	74.6	2.9	4.8	3.2	1.5	7.8	5.2
Group 3	*B. stejnegeri*	16.0	59.2	2.7	7.9	1.1	4.1	8.9
	*K. koreana*	22.2	0.0	18.6	29.4	0.0	29.6	0.2
	*O. koreanus*	42.8	41.0	4.1	0.9	0.9	2.3	8.0
	* R. huanrenensis*	57.2	21.9	3.3	2.1	3.4	7.2	5.0

**Table 4 animals-11-02185-t004:** Estimating the suitable habitat area of amphibians per habitat characteristics. (unit: km^2^).

Group	Scientific Name	Current	RCP 4.5			PHL * (%)	RCP 8.5			PHL * (%)
			2030	2050	2080		2030	2050	2080	
Group 1	*H. japonica*	69,191	74,273	50,748	23,119	66.6	52,626	41,463	22,798	67.1
	*K. borealis*	24,392	30,814	39,835	61,206	−150.9	48,429	33,656	41,009	−68.1
	*G. rugosa*	60,730	52,518	41,887	19,256	68.3	35,123	34,411	31,959	47.4
	*L. catesbeianus*	28,523	62,436	37,986	27,067	5.1	40,263	55,541	42,311	−48.3
	*P. chosenicus*	10,901	4298	298	2000	81.7	3255	412	168	98.5
	*P. nigromaculatus*	65,551	71,358	49,658	28,648	56.3	57,881	44,831	19,902	69.6
	* R. coreana*	47,791	33,675	17,128	15,420	67.7	28,215	25,815	16,617	65.2
Group 2	*B. gargarizans*	65,144	62,370	41,381	21,368	67.2	46,998	48,268	24,268	62.7
	*B. orientalis*	56,561	47,634	30,044	2529	95.5	25,427	30,774	3354	94.1
	*H. leechii*	68,673	38,858	22,711	11,747	82.9	24,299	31,160	15,452	77.5
	*H. quelpaertensis*	1585	1427	888	1510	4.7	1343	253	1220	23.0
	* R. uenoi*	66,419	13,369	9417	2261	96.6	13,130	7152	2356	96.5
Group 3	*B. stejnegeri*	17,160	16,158	4288	265	98.5	3916	1448	59	99.7
	*O. koreanus*	21,351	9675	5193	330	98.5	4920	2953	118	99.4
	*K. koreana*	45,627	76,576	84,130	87,674	−92.2	79,896	78,035	87,252	−91.2
	* R. huanrenensis*	31,061	13,481	6600	670	97.8	6466	3026	1228	96.0
Total area	100,411 ^a^									

^a^ Source: Ministry of Land Infrastructure and Transport, Republic of Korea, 2018. * PHL: Potential habitat loss rate = (current–2080)/current × 100. The minus values of this estimation indicate increasing the potential habitat by 2080.

**Table 5 animals-11-02185-t005:** Basic statistics for the species richness of amphibians for different habitat characteristics in South Korea.

**RCP**	**Year**	**All**		**Group 1**		**Group 2**		**Group 3**	
		**Avg**	**Max**	**Avg**	**Max**	**Avg**	**Max**	**Avg**	**Max**
	Current	7.33	12.00	2.60	6.00	3.48	5.00	1.25	4.00
4.5	2030	6.19	14.00	2.94	6.00	2.01	5.00	1.23	4.00
	2050	4.71	14.00	2.28	6.00	1.39	5.00	1.04	4.00
	2080	3.71	12.00	2.35	6.00	0.45	5.00	0.92	4.00
8.5	2030	5.01	13.00	2.69	6.00	1.31	5.00	1.02	4.00
	2050	5.00	13.00	2.92	6.00	1.19	5.00	0.88	4.00
	2080	4.30	11.00	2.79	6.00	0.6	5.00	0.91	4.00

**Table 6 animals-11-02185-t006:** Areas of the species richness classes of amphibians in South Korea from different habitat groups (unit: km^2^).

Habitat			RCP4.5			RCP8.5		
Groups	SR * Classes	Current	2030	2050	2080	2030	2050	2080
	Low (0–1)	23,489	15,970	34,572	51,231	29,348	33,026	52,852
Group 1	Mid (2–4)	44,823	52,094	48,351	35,381	47,418	48,409	34,879
	High (5~7)	27,802	28,220	13,361	9672	19,505	14,838	8542
	Low (0–1)	22,604	41,113	62,642	89,066	63,930	59,266	86,693
Group 2	Mid (2–3)	36,568	49,692	29,856	7098	27,614	33,604	9359
	High (4~5)	36,942	5479	3786	120	4727	3403	221
	Low (0–1)	65,259	72,324	86,187	95,317	86,137	92,280	95,001
Group 3	Mid (2–3)	25,866	19,700	9125	909	9458	3625	1262
	High (4)	4989	4260	972	58	676	368	10
All	Low (0–4)	13,785	22,241	50,037	73,827	43,616	49,586	70,203
	Mid (5–8)	68,190	67,069	44,224	21,996	50,803	45,506	25,945
	High (9~14)	14,139	6974	2023	461	1852	1181	125

* SR: Species Richness.

## Data Availability

All of the occurrence data in this study are available on the EcoBank official website (https://www.nie-ecobank.kr/cmmn/Index.do?lang=en, accessed on 20 July 2021). EcoBank is the first ecological database platform in Korea. For more information, please read: Kim, H.W.; Yoon, S.; Kim, M.; Shin, M.; Yoon, H.; Kim, K. EcoBank: A flexible database platform for sharing ecological data. *Biodiversity Data Journal*
**2021**, *9*, e61866, doi:10.3897/BDJ.9.e61866.

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
