# Peer review of "Potential Distribution of Amphibians with Different Habitat Characteristics in Response to Climate Change in South Korea"

_animals, 2021, doi:10.3390/ani11082185_

Round 1

Reviewer 1 Report

English writing still is in need of improvement.

Author Response

Thank you for your suggestions and comments so far. According to your suggestions on improving English writing, we requested this job to Editage, an English Editing company. We marked up all of our revisions on the manuscripts through the 'Track Change' function in MS Word. 

Please take a look one more time, and let us know if you still feel the improvement of English writing of our manuscript is not yet enough. 

Sincerely, 

Hyun Woo Kim 

Reviewer 2 Report

The authors have improved the ms adressing thoroughly my concerns with the previous version.

Author Response

Thank you very much for your valuable suggestions. Your ideas have been very useful in improving the quality of our manuscript. 

According to a suggestion by another reviewer on improving English writing, we requested this job to Editage, an English Editing company (www.editage.com). We marked up all of our revisions on the manuscripts through the 'Track Change' function in MS Word. 

Please take a look one more time, and let us know if you have any additional suggestions and comments. 

Sincerely, 

Hyun Woo Kim 

Reviewer 3 Report

The authors have done a good job responding to my earlier comments.

Author Response

Thank you very much for your valuable suggestions. Your ideas have been very useful in improving the quality of our manuscript. 

According to a suggestion by another reviewer on improving English writing, we requested this job to Editage, an English Editing company (www.editage.com). We marked up all of our revisions on the manuscripts through the 'Track Change' function in MS Word. 

Please take a look one more time, and let us know if you have any additional suggestions and comments. 

Sincerely, 

Hyun Woo Kim 

This manuscript is a resubmission of an earlier submission. The following is a list of the peer review reports and author responses from that submission.

Round 1

Reviewer 1 Report

I praise the authors for the work conducted here, and I highlight some potential improvements. Please find the minor correction on the file attached, I have stopped highlighting small mistakes half way through the manuscript because there were too many of them.

Major points:

- The list of species is not up to date: some newly described species are missing and some of the taxonomy is outdated.

- Instead of excluding some of the species because of the lack data, the authors should rely of other datasets, such as GBIF

- The range maps of some species are wrong. Hynobius quelpaertensis is not restricted to Jeju island.

- Regarding the variables used for the models, the inclusion of other biotic variables, such as NDVI, would provide a better resolution to the question asked in this study

Author Response

Thank you for your valuable comments including your minor comments directly on the *.pdf file. We forgot to indicate the revisions for your minor suggestions, but we have corrected them as far as possible.  If you still find uncorrected comments, just let us know again. 

We attached a file related to our corrections for your major points. Please take a look, and let us know if you still have questions. 

Best Regards, 

Hyun Woo 

Reviewer 2 Report

This study models the potential distribution of Korean amphibians in response to climate change. The authors apply established methods, and discuss the results carefully, pointing out the limitations of this approach.

I find the focus on the three groups of species not convincing. There is considerable heterogeneity within these groups, and to analyse species richness at the level of these groups (which contain four to seven species) does not appear to be highly informative. Instead, the discussion could focus on the factors (or combination of variables) that cause the predicted changes in distribution of single species (this is done shortly for a few species only).

In the summary and abstract, the use of these results for implementation of "conservation strategies" is stressed. In the main body of the paper, however, no detailed explanations on how these findings may help to design conservation measures are given.

The English is generally clear, but some improvement is recommended. The frequent typographical flaws, particularly concerning species names, are a nuisance.

A few minor comments:

line 33: "All species investigated should lose their suitable habitat, except ..." might be misunderstood that extinction of 13 species is predicted.

line 45: unclear what "climate change ... stable" means (perhaps drop "change")

line 53: not all amphibians require humid habitats with low temperatures

line 114: genus names should only be abbreviated after the full name has been mentioned once

line 165: "higher ... superiority" ??

line 229: a low rate of habitat loss should be good news for a species; rephrase sentence ?

line 231: meaning not quite clear (perhaps a "not" missing?)

line 386: "current ecological niches" seems to be used in the meaning of "suitable habitats" here, which I think is fundamentally wrong. The whole study is based on an analysis of the currently realized niches (in terms of bioclimatic factors), and on the assumption that these niches (organism-habitat relation) stay constant. The instances of suitable conditions are predicted to decline, but not the ecological niches as such.

Author Response

Thank you for your valuable suggestions. We have revised as far as we can according to your comments. Please take a look again, and let us know if you have any additional comments. 

Best Regards,

Hyun Woo Kim

Reviewer 3 Report

I noticed three main issues:

1) The ms suffers still from a remarkable number of formatting issues. Do authors not care for reading ms before submission? I would have returned the ms for fixing these issues before sending to reviewers!

2) The grouping of Korean amphibian species to three "guilds" is not plausible at the moment. The rationale for these groups mut be better worked out.

3) Many of the conclusions are trivial and would not have needed a complex modelling process to find out. Some parts of the model predictions are interesting, but are hidden within the more trivial parts.

In conclusion, I am not very impressed from the ms. Issues 2 and 3 should be worked out better. Its originality is rather low, it focusses on the modelling approach and not really on the ecology/life-history of the amphibians treated. So it is merely a formalization of what most amphibian ecologists would have concluding from the known ecology of the species. I believe that the impact of this ms on the community will be rather low. .

Author Response

Thank you for your valuable suggestions and comments. We have revised our previous manuscript as far as we can. Please take a look again, and let us know if you have any additional suggestions. 

Sincerely, 

Hyun Woo Kim 

Reviewer 4 Report

This manuscript explores the distribution and predicted distribution of amphibians in South Korea with respect to climate change scenarios. Understanding the potential effects of climate change on organisms and their distributions is important. However, it is not clear what this particular analysis brings to the table that makes it of interest to a broad readership. Below I outline my concerns and suggestions.

Major concerns and suggestions:

1) Throughout the manuscript the authors categorize the amphibians into 3 groups based on supposed similarities in life history or ecology. Unfortunately the authors do not provide much in the way of justification or rationale for this beyond a vague statement in lines 139-140. I would argue that the focus on these groups masks some of the important differences among species, even within “groups”. I would prefer it if the authors looked at variation among species rather than among groups – this would be more useful and informative.

2) The authors do not really provide a compelling argument as to why this particular study is of import to a broader readership outside of South Korea, and how it contributes to a broader understanding of climate change effects on amphibians. In general the predicted changes are enumerated but the manuscript does not really explore why the patterns observed are seen.

3) The results are very difficult to follow and are not effectively or efficiently communicated. I think the authors should look at how they might more effectively communicate their results, especially with regards to species and the variations among species.

Other concerns and suggestions (given in order observed in manuscript):

line 34 and throughout: Lithobates catesbeianus are not native to South Korea. This should be made clear in the Abstract and it should be identified as such before the Discussion. It is also not clear if only its range in South Korea was used to predict or whether its native range was.

lines 45-46: When was this assumption made? It has not been assumed for quite some time.

lines 48-49: Not all species will be negatively effected. Many will but this is not universal.

lines 69-79: This paragraph does not do a good job of justifying this study. Why is this important?

line 80: Insert “potential” before “effects”

line 96: I think you mean “fauna” not “flora” – “flora” typically refers to plants and fauna is animals.

lines 98-105: Citations for these?

lines 122-123: At what scales?

line 133: To be consistent with the other time intervals why not do 1991-2000?

line 184 and throughout: Species names need to be italicized.

lines 186-188: This suggests to me that the model is not very good for several species and so raises questions about the conclusions being drawn.

lines 189-191: This sentence does not make sense.

Tables: I don’t see any tables in the materials provided in the pdf for review.

lines 273-274: Why average? Also what was averaged?

lines 275-276: Why were these categories decided on?

line 391: Don’t other groups also share their habitat with humans?

lines 424-428: SO how determine where is better?

lines 430-435: Why do you need to predict current distributions? Do these predictions match current distributions?

line 480: This statement ignores the variability among species within groups which is as great as variation among groups.

Author Response

Thank you for your valuable suggestions and comments. Based on them, we have revised our manuscript as far as possible. please take a look one more time, and let us know if you have any further opinions. 

Sincerely, 

Hyun Woo Kim 
